biomimetics/biomechanics

human foot structure, transverse tarsal joint, oblique axis, free moment, constructive approach

**Author for correspondence:**
Tsung-Yuan Chen
e-mail: chen.tsungyuan@arl.sys.es.osaka-u.ac.jp

# Free moment induced by oblique transverse tarsal joint: investigation by constructive approach

Tsung-Yuan Chen[1], Takahiro Kawakami[1], Naomichi Ogihara[2] and Koh Hosoda[1]

[1]Graduate School of Engineering Science, Osaka University, Osaka, Japan
[2]Department of Biological Sciences, The University of Tokyo, Tokyo, Japan

T-YC, 0000-0002-1155-1372

The human foot provides numerous functions that let humans deal with various environments. Recently, study of the structure of the human foot and adjustment of an appropriate reaction force and vertical free moment during bipedal locomotion has gained attention. However, little is known about the mechanical (morphological) contribution of the foot structure to the reaction force and free moment. It is difficult to conduct a comparative experiment to investigate the contribution systematically by using conventional methods with human and cadaver foot experiments. This study focuses on the oblique transverse tarsal joint (TTJ) of the human foot, whose mechanical structure can generate appropriate free moments. We conduct comparative experiments with a rigid foot, a non-oblique joint foot (i.e. mimicking only the flexion/ extension of the midfoot), and an oblique joint foot. Axial loading and walking experiments were conducted with these feet. The axial loading experiment demonstrated that the oblique foot generated free moment in the direction of internal rotation, as observed in the human foot. The walking experiment showed that the magnitude of the free moment generated with the oblique foot is significantly lower than that with the rigid foot during the stance phase. Using this constructive approach, the present study demonstrated that the oblique axis of the TTJ can mechanically generate free moments. This capacity might affect the transverse motion of bipedal walking.

## 1. Introduction

The complex structure of a human foot has numerous important features for dealing with various environments during bipedal

locomotion [1]. The intricate foot structure (i.e. joints, bones, muscles and ligaments) facilitates shock absorption [2], adaptation to uneven terrain, balancing [3] and leverage for propulsion [4]. These functions suggest that the foot structure plays an important role in adjusting the appropriate ground reaction force/moment during bipedal locomotion. For example, the foot structure and its movements are associated with the ground reaction force peaks in different phases of the locomotion [3]. The different human arch heights affect the generation of the ground reaction force [5]. The external rotation of the foot occurs with significantly greater free moment peaks and relative net impulses compared with normal walking [6]. The pronation of foot generates a higher free moment peak during bipedal locomotion [7]. A cadaver experiment demonstrated that the internal vertical free moment was generated by the talus rotation [8]. However, the ground reaction force and vertical free moment generation by the underlying human foot structural deformation needs to be further explored.

This study focuses on free moment (FM) generation by the oblique transverse tarsal joint (TTJ) which is also called the Chopart joint in clinical research. Some studies found that the vertical component of foot structural movements (i.e. abduction/adduction) significantly affects the free moment generation [6–8]. The free moment is a torque that is applied at the vertical axis originating at the foot's centre of pressure (CoP) [9,10]. It is one of the essential components that can change the condition of the rotation of the body about the vertical axis during walking [11]. It has been used as a tibial stress fracture predictor in clinical applications [12] and is considered a useful biomechanical indicator in static postural [13], arm-swing [11,14] and gait analyses [10]. It also contributes to suppressing whole-body oscillation and compensating insufficient moment cancellations between the upper and lower body during the stance phase [15].

The TTJ is the joint located between the hindfoot and forefoot, and it contributes significantly to the structural deformation of the human foot. Numerous studies assume that the TTJ is a non-oblique joint in biomechanics studies, such as truss mechanism [16–18]. However, TTJ is not exactly the same as their assumption. The early works of Elftman, Manter and Hicks suggested that the TTJ moves about two axes of motion, the longitudinal and oblique axes [19–21]. The movement of the longitudinal axis generates the inversion/eversion of the feet, and the oblique axis generates combinations of abduction/adduction and dorsiflexion/plantarflexion [18,22–24]. In addition, following the measurement technique updates, recent studies suggested that the TTJ has a single tri-planar axis of motion [24,25]. Nester *et al*. [25] suggested that the oblique TTJ can invert, adduct, and dorsiflex between the heel strike and forefoot loading, and it everts, abducts, and plantarflexes after heel-off. This structure is so complex and difficult to directly measure that the information is relatively sparse [24].

To sum up these reports, we hypothesize that a single oblique joint structure has the potential to approach the TTJ and affect the free moments generation during locomotion. Nevertheless, little is known about this because it is difficult to conduct quantitative and comparative experiments with/without an oblique joint structure by conventional approaches such as cadavers and human experiments.

In recent years, a cross-disciplinary research method based on the embodiment concept [26], also called as constructive approach, has gained to prominence. Hashimoto *et al*. built a robotic foot with the medial longitudinal arch of the foot to demonstrate that this foot structure has an impact absorbing function during bipedal locomotion [27]. Narioka *et al*. built a robotic foot with a human-like ankle-foot complex and found that truss and windlass mechanisms play important roles in shock absorption, energy storage and reuse. They also found that truss and windlass mechanisms work effectively with an appropriate tone of the plantar aponeurosis [17]. Kawakami and Hosoda built a robotic foot with an oblique midfoot joint and found that this oblique joint was able to stabilize the body rolling posture during bipedal walking [28]. Venkadesan *et al*. used a simple mechanical transverse arch model to investigate the relationship between stiffness and geometry of the TTJ [29]. These studies suggest that the constructive approach method has great potential towards conducting quantitative and comparative experiments with/without the oblique joint structure and measuring the resulting free moment generation.

In this study, we adopted a constructive approach in which we built a robotic foot to demonstrate the function of the foot's structural mechanisms. We constructed three robotic feet: a rigid foot, one foot with a non-oblique joint and one with an oblique axis of TTJ. We conducted two experiments: an axial loading experiment and a walking experiment. An axial loading machine and a musculoskeletal bipedal walking robot were built to conduct the experiments.

The rest of this paper is organized as follows. First, we introduce the experimental preparation: three kinds of robotic feet, with rigid, non-oblique and oblique TTJ; a loading apparatus for the axial loading experiment; and a biped robot for the walking experiment in the section of the experimental methods. Then, the results of the two experiments are presented in the Results section and discussed with regards to the ability of the foot structure in the Discussion section.

# 2. Experimental methods

## 2.1. Conditions

To conduct the comparative experiments, we built a rigid foot, a foot equipped with a non-oblique joint and a foot with an oblique joint. The detailed design purpose conditions of each foot are shown below.

— Condition (a): Foot with a non-oblique joint allowing flexion/extension.
— Condition (b): Rigid foot without a midfoot joint structure.
— Condition (c): Foot with a human-like oblique joint representing the TTJ.

Figure 1 illustrates the test robotic feet. The rigid foot is built using an oblique foot with an aluminium plate. This plate constrains the rotational degree of freedom about the oblique joint. The rigid foot has only one talocrural joint allowing for dorsiflexion/plantarflexion. It is also used as the comparison standard. The non-oblique joint foot is equipped with a hinge joint allowing for flexion/extension and two McKibben-type pneumatic artificial muscles (PAMs) [30] as elastic plantar fascia. The pressure and size of its PAMs are set to the same values as that the oblique foot. The plantar surface boundary condition of the tested robotic feet is sliding.

## 2.2. Human-mimicking oblique foot

Anatomical investigations have reported that the oblique axis (the left side of figure 2) of the TTJ is positioned approximately 57° medial to the sagittal plane and 52° superior to the transverse plane [19–21]. Based on a report by Nester *et al.* [25], we hypothesize that the single oblique joint structure is able to approximate the transverse tarsal joint. In this study, we call this the oblique transverse tarsal joint. The developed human mimicking oblique foot is equipped with a functionally equivalent oblique TTJ.

Figure 3 illustrates the developed oblique joint robotic foot and shows the corresponding relationship between the human oblique axis and the oblique robotic foot. This robotic foot is equipped with an oblique TTJ and a truss mechanism [17]. It is 200 mm long and 80 mm wide, while the calcaneus is 56 mm wide. It has two joints: the talocrural joint and the oblique axis of the TTJ. We focused on the effects of the oblique TTJ structure. This robotic foot is not equipped with toes. With this oblique joint, the foot is able to move in a tri-planar motion. This could simplify the complexity of the robotic foot and provide a clear view of the effects of the oblique axis structure.

This foot is also equipped with two McKibben-type PAMs (see figure 4) on the bottom to mimic the elastic plantar fascia. The diameter and length of the inner tube of PAMs are 7 and 120 mm, respectively. The pressure of the PAMs is set to 500 kPa. The stiffness is linearly related to the pressure [30]. We decided the pressure parameter by integrating the investigation results of the elastic properties of the plantar fascia [31] and the robotic studies [17,32]. With the set-up, the stiffness of the single pneumatic muscle is approximately $9.8 \text{ N mm}^{-1}$.

## 2.3. Free moment measurement

In this study, we used a force plate (Tec Gihan Co., Ltd, TF-3040) to measure the free moment in the axial loading and walking experiments. The experiments were conducted using the left foot, and the measurement coordinate system is shown in figures 5 and 8. The positive and negative magnitudes of the measured free moment represent the external and internal free moments, respectively. The free moment is calculated using formula (2.1), where FM and $Mz$ denote the free moment and the vertical moment at the centre of the force plate, respectively. $Fx$ and $Fy$ are the $x$- and $y$-axis components of the ground reaction force, respectively, and $CoPx$ and $CoPy$ are the $x$- and $y$-axis components at the CoP on the force plate, respectively. The sampling rate of the force plate was 1 kHz.

$$\text{FM} = Mz - Fy(CoPx) + Fx(CoPy) \tag{2.1}$$

## 2.4. Axial loading experiment

The axial loading experiment is widely used in the ground reaction force/moment analysis of the cadaver feet [8,33,34]. In this study, we built a vertical load examination machine (figure 5) to measure the free moment generations of the test feet. The loading machine is equipped with a pneumatic cylinder (SMC CQ2KB63-50DZ) as a vertical load generator and a connector to facilitate

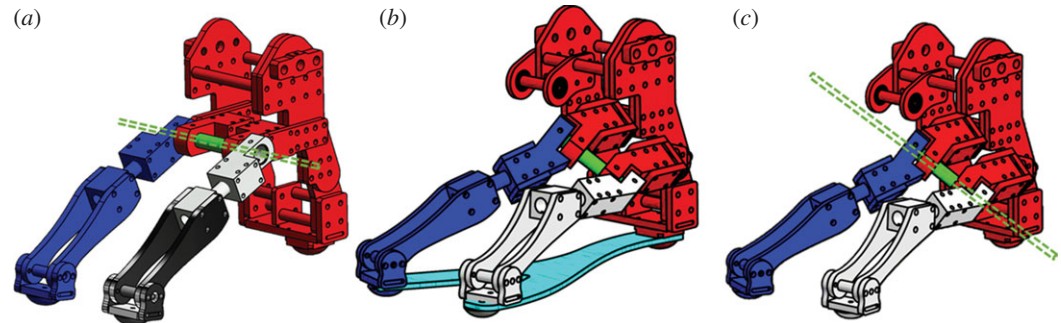

**Figure 1.** Test robotic feet: (*a*) non-oblique joint foot, (*b*) rigid foot, (*c*) oblique joint foot.

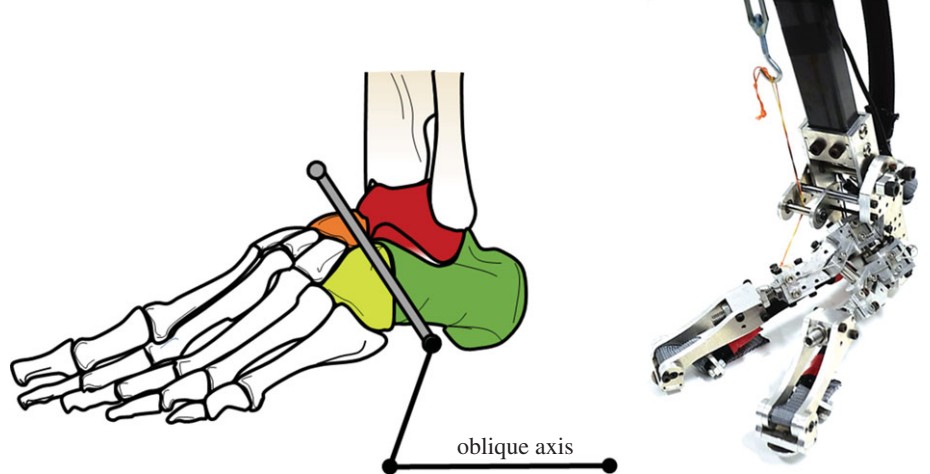

**Figure 2.** Oblique axis of transverse tarsal joint and mimicking oblique robotic foot (left foot). The grey line represents the human oblique axis is located between the calcaneus (dark green) and cuboid (light green), and the talus (red) and navicular (orange).

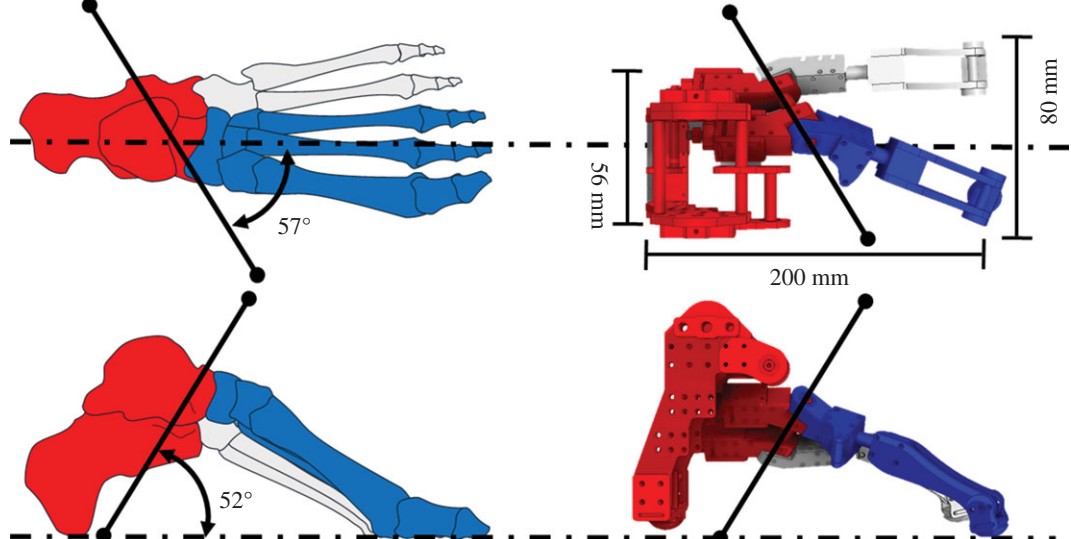

**Figure 3.** Corresponding positions of human and oblique robotic feet.

switching of the test feet. To prevent the damage from the test feet, a foam mat made of urethane was set up on the force plate (the static and kinetic friction coefficients were approximately 0.3 and 0.18, respectively). The pressure of the cylinder was controlled with a proportional directional control valve (FESTO MPYE-5-M5-010-B-SA), an Arduino Duo microcontroller and an amplifier circuit. A force plate was installed at the bottom of the loading machine to measure the free moment. To compare the

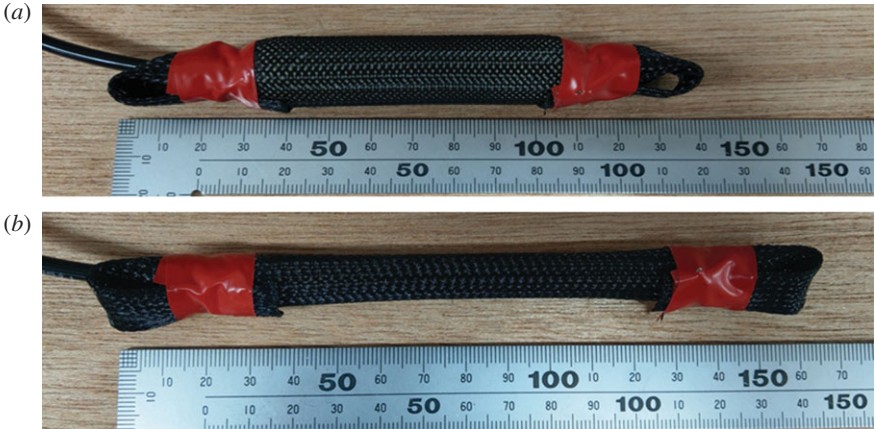

**Figure 4.** McKibben pneumatic artificial muscles. (*a*) Pressurize status, (*b*) exhaust status.

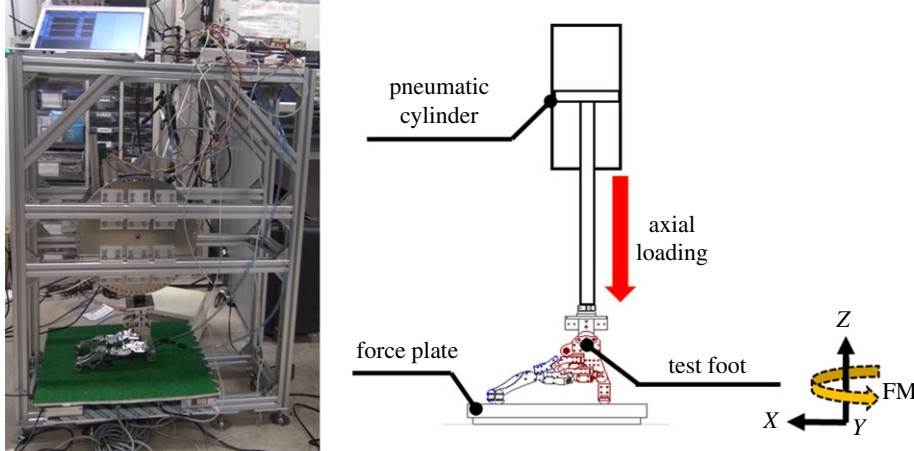

**Figure 5.** Axial loading machine.

cadaver experimental results [8] and accommodate the weight of our designed walking robot, the vertical load was determined to be 150 N. In this experiment, each test foot was examined over five trials.

## 2.5. Biped walking robot experiment

The purpose of this experiment was to examine the performance of the test feet on a biped walking robot. We measured the free moment during the stance phase of the developed biped walking robot. In the experiment, we installed the rigid, non-oblique and oblique feet on the developed musculoskeletal biped walking robot. Figure 6 illustrates the musculoskeletal biped walking robot and its muscle set-up. The details of the muscle arrangement are listed in table 1. This musculoskeletal biped walking robot that weighed 8.3 kg, was 1.1 m tall, and 0.25 m wide, with a length of 0.65m. Each leg had hip, knee, and ankle joints (i.e. the talocrural joints of the test robotic feet), which were actuated by antagonistic movements [35]. To clearly observe the contributions of the test feet, the hip and knee joints of the walking robot are only allowed for flexion/extension. To simplify the control sequence and implement antagonistic movements, only the gluteus maximus, vastus lateralis and soleus in this robot were equipped with PAMs. Other muscles were equipped with springs. The diameter of the inner tube of the PAMs was 10 mm. The lengths of the gluteus maximus, vastus lateralis and soleus were 150, 210 and 230 mm, and the stiffness of the tibialis anterior, iliacus and gastrocnemius were 2.1, 2.1 and 19.6 N mm$^{-1}$, respectively.

We designed a walking pattern (figure 7) sequence, based on the investigations of the activation of the human muscles during bipedal walking [36] and previous musculoskeletal biped walking robotic research [17,28], and tuned it through empirical testing that made sure the robot can walk over three steps without falling. This walking pattern was burned into an Arduino Duo microcontroller with a

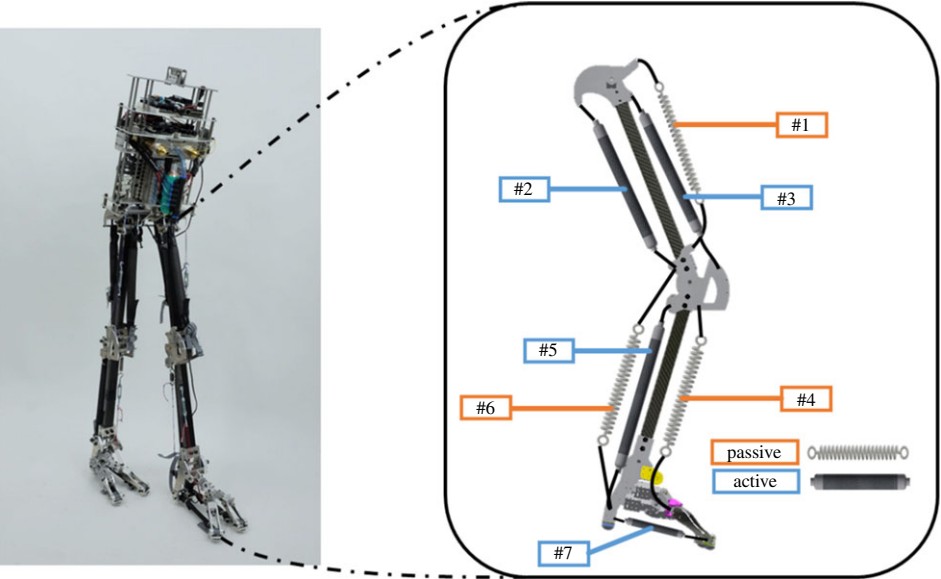

**Figure 6.** Musculoskeletal biped walking robot and muscle set-up.

**Table 1.** Arrangement of the PAMs.

| no. | muscle name | function | drive type | muscle type |
|---|---|---|---|---|
| 1 | iliacus | hip flexion | passive | mono |
| 2 | gluteus maximus | hip extension | active | mono |
| 3 | vastus lateralis | knee extension | active | mono |
| 4 | tibialis anterior | ankle flexion | passive | mono |
| 5 | soleus | ankle extension | active | mono |
| 6 | gastrocnemius | knee flexion, ankle extension | passive | bi |
| 7 | plantar fascia (x2) | elevation arch | active | mono |

custom signal amplifier board to control the activations of the pneumatic solenoid valves and PAMs. Both legs were actuated by the same control sequence in an alternating pattern. A load cell (Tec Gihan Co., Ltd, USL06-H5-500N) was installed at the heel of the feet to detect the heel contact and determine the switching. With these settings, the robot's average walking speed was 0.5 m s$^{-1}$, and it ensured that the robot could walk three steps continuously without falling.

The experimental environment and procedure are shown in figures 8 and 9. The robot was initially held by the experimenter and the left leg is raised approximately 20° and subsequently released. It stepped forward and made contact with the force plate. The free moment during the stance phase was measured using the force plate. Each case was examined over five trials, and the average values were calculated to represent the results. As the robotic walking is difficult to completely start from the same initial position, the free moment was measured at non-steady-state walking.

## 2.6. Statistical method and data analysis

All the kinematic data were filtered using a fourth-order low-pass Butterworth filter with a 15 Hz cut-off frequency, and the statistical test was conducted using the one-way ANOVA with Tukey's post HSD (honestly significant difference) tests. The filter was programmed using Python and integrated the Scipy and Pandas libraries. The statistical test was processed using Microsoft Excel.

# 3. Results

The axial loading experimental results are shown in figure 10, and the CoP positions are listed in table 2. The average value was calculated by measurements across five experiments on each foot. The statistical

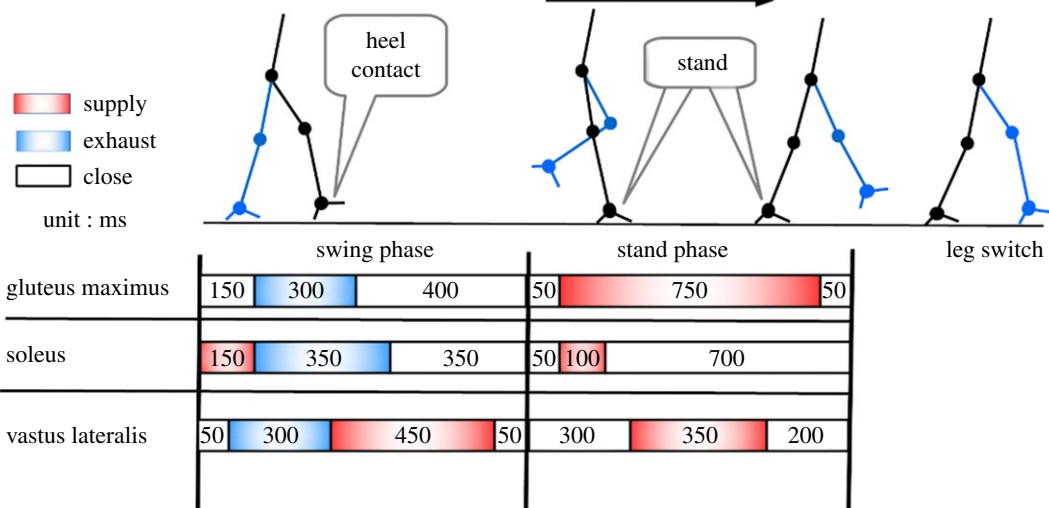

**Figure 7.** The single leg walking pattern.

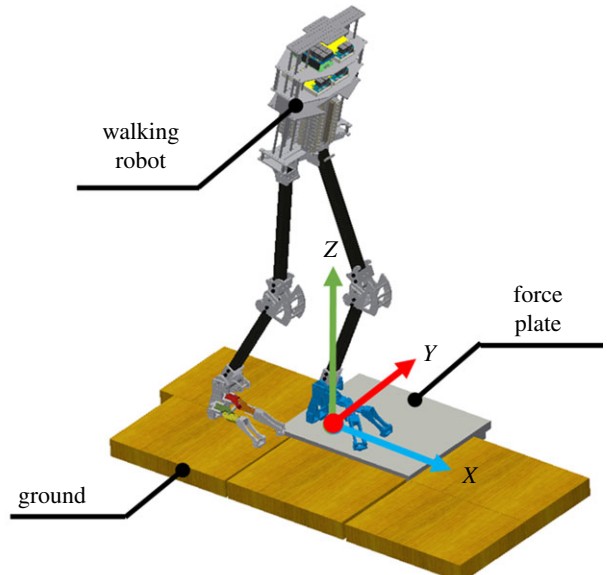

**Figure 8.** Environment for walking experiment.

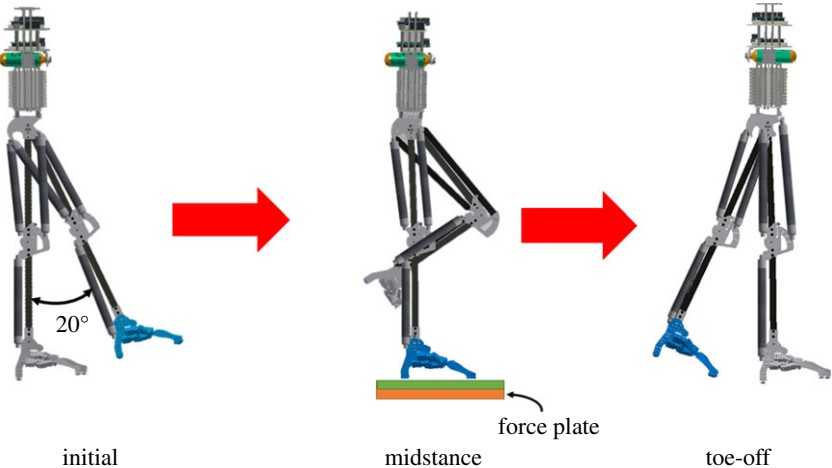

**Figure 9.** Walking procedure.

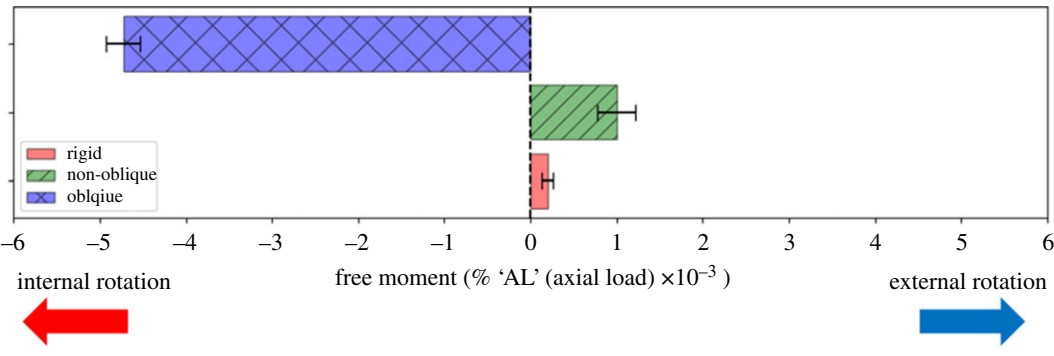

**Figure 10.** Average values of free moment generation under 150 N axial load. (The magnitude is normalized by the axial load (AL).)

**Table 2.** The centre of pressure (CoP) measurement of the axial loading experiment.

| condition | CoP-X (mm) | CoP-Y (mm) | HCP-X (mm) | HCP-Y (mm) |
|---|---|---|---|---|
| non-oblique foot | −42.4 ± 0.1 | 12.4 ± 0.8 | −0.12 ± 0.04 | −0.02 ± 0.04 |
| oblique foot | −20.6 ± 0.1 | −7.1 ± 0.5 | 0.22 ± 0.16 | 0.1 ± 0.07 |
| rigid foot | −35.8 ± 0.2 | −7.9 ± 2.4 | 0.18 ± 0.08 | 0.16 ± 0.09 |

analysis was performed to assess the difference between the average magnitude of the free moment. The levels of significance between the test feet were all less than 0.01 ($p$-value < 0.01). The results indicate that only a few external free moments were passively generated with the non-oblique and rigid feet. Therefore, they might be able to be considered to have generated almost zero free moment. The average normalized values were $0.196 \pm 0.063$ and $1.0 \pm 0.223$ ( mean ± s.d. $\times 10^{-3}$). By contrast, the free moment was generated in the direction of internal rotation for the oblique foot. The average normalized value was $-4.729 \pm 0.198$. Compared to the results of the rigid foot, the oblique foot exhibited a larger internal free moment.

The free moment during the bipedal robot walking is shown at the figure 11. The occurred timing of midstance and toe-off in each condition was different. The midstance phases of rigid, non-oblique, and oblique feet occurred approximately at 38.4%, 41.4% and 38.8% of stance phase, respectively, and the toe-off of the rigid, non-oblique, and oblique feet occurred approximately at 77.3%, 73.5% and 78.5% of stance phase, respectively.

The normalized negative average peak free moment (figure 12$a$) of the rigid, non-oblique, and oblique feet were $-2.554 \pm 0.313$, $-1.636 \pm 0.285$ and $-1.287 \pm 0.498$ (mean ± s.d. $\times 10^{-2}$), respectively. The values for the non-oblique and oblique feet were 35.94% and 49.59% lower than that of the rigid foot, respectively.

The normalized average values of the measured positive peak free moment (figure 12$b$) of the rigid, non-oblique, and oblique feet were $2.796 \pm 0.344$, $2.304 \pm 0.336$ and $1.752 \pm 0.192$, respectively. The values for the non-oblique and oblique feet were 17.6% and 37.34% lower than that of the rigid foot, respectively.

The average value of peak to peak change (figure 12$c$) showed that the oblique foot ($3.039 \pm 0.414$) was lower than the non-oblique foot, and lower ($3.94 \pm 0.272$) than the rigid foot ($5.35 \pm 0.6$). The values for the non-oblique and oblique feet were 26.36% and 43.19% lower than that of the rigid foot, respectively.

## 4. Discussion

This present study investigated the capacity of the foot to generate a free moment during axial loading and walking using a robotic foot with and without an oblique transverse tarsal joint.

In the axial loading experiment, a significant internal free moment was generated with the oblique foot than with the rigid foot or non-oblique foot while bearing a constant axial load. By observing the movement of the oblique foot, the forefoot tends to externally rotate while bearing the axial loading. Therefore, this movement may induce friction to generate the resisting internal free moment. The forefoot motion was similar to the cadaver finding by Ito *et al.* [33]. Note that the direction of the larger free moment is the same as, and the value is close to the experimental result of the cadaver foot axial loading experiment by Seki *et al.* [8] (their dimensionless valid value is $-3.69 \times 10^{-3}$).

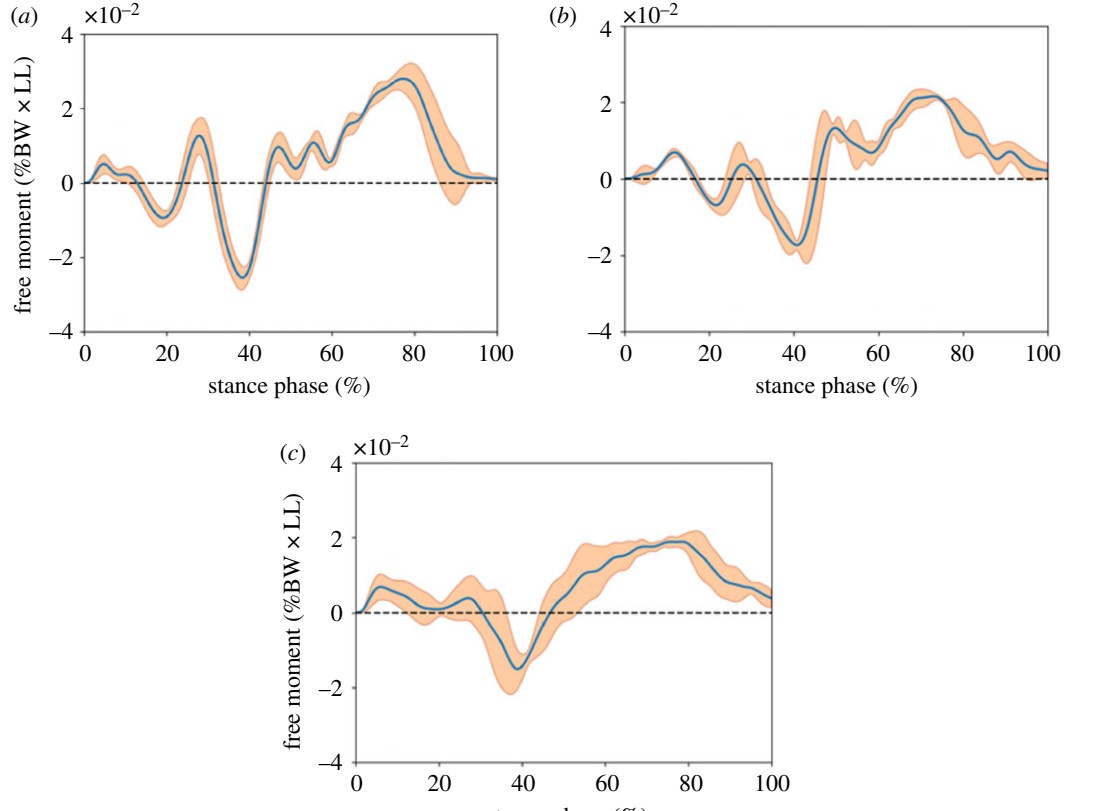

**Figure 11.** Free moment during the robot walking. (The blue line is the average; the orange shadows are standard deviation variability bands. The magnitude is normalized by the robot's body weight and leg length. BW and LL represent the robot's body weight and leg length, respectively).

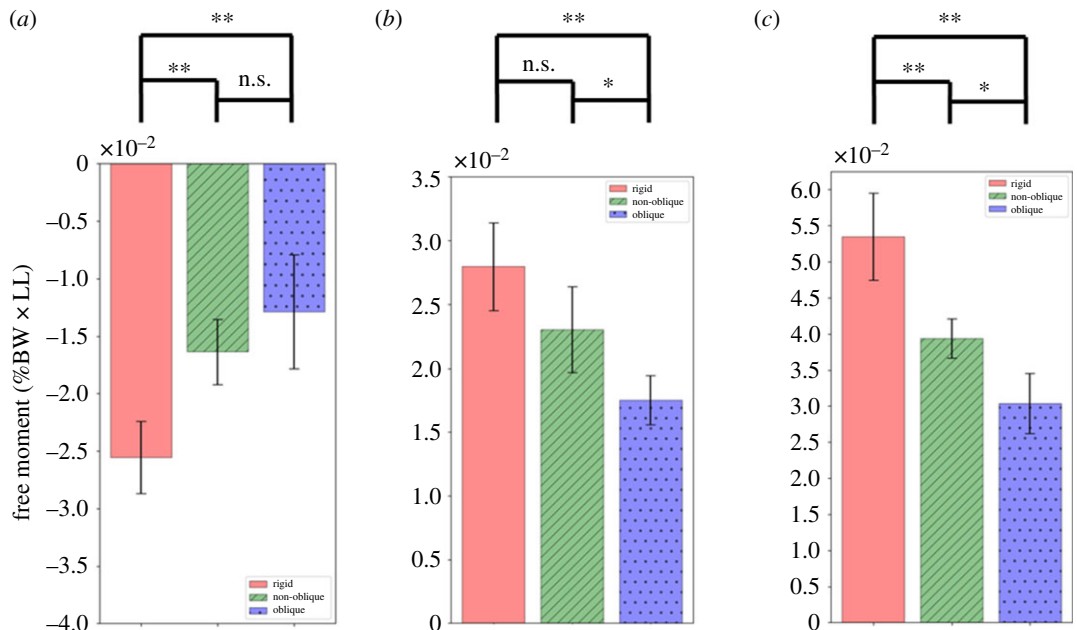

**Figure 12.** Peaks of the moment during the stance phase (**, * and n.s represent $p < 0.01$, $p < 0.05$ and not significant, respectively). (*a*) Negative peak, (*b*) positive peak, (*c*) peak to peak.

In the walking experiment, the free moments of all three conditions exhibit a biphasic shape, which produced the negative free moment in the first half of the stance phase and produced the external free moment in the latter half. [6,10]. Comparing the human waveform, we found that the oblique foot's

waveform was very similar to the human result. The smallest amplitude and peaks of the free moment occurred with the oblique foot. The positive peak (i.e. external rotational peak) of the oblique foot was statistically significantly smaller than that of the non-oblique and rigid foot. Figure 11 shows the waveforms of the free moment; we observe that all the test feet showed significant valleys in the first half. In the latter half (i.e. the duration approx. 38% to 78% of the stance phase), the oblique foot significantly displayed the smoothest results in comparison to the rigid and non-oblique feet. By observing the movement of the oblique foot while bearing the pressing force (the left foot movement is shown in [37]), we found that the ankle part of the oblique foot has partial internal rotation in the transverse plane. It seems that the transient internal rotational movement may reduce the contrary vertical rotation generated by the swing leg and makes the free moment of the latter part of the oblique foot smaller than the rigid foot. However, this hypothesis still needs detailed research to confirm.

Considering the applications of the findings of this study. The proposed oblique robotic foot has a high potential to be applied in the design of the yaw moment compensator of the bipedal walking robot and improve the walking performance [38–40]. These renewed findings also provide the information to better understand the TTJ in the biomechanics study.

We constructed a robotic foot with an oblique transverse tarsal joint to demonstrate its contribution to free moment generation. The oblique foot generated a free moment in the direction of the internal rotation, as observed in the human foot, when an axial loading was applied. In the bipedal walking experiment, the magnitude of the free moment during the stance phase of the gait was lowest when the robot walked using the foot with the oblique joint. Most especially, the external free moment was significantly reduced with the oblique foot. These results suggest that the foot mechanism with the oblique transverse tarsal joint might affect the transverse motion of bipedal walking and highlights the importance of the morphology of the human foot.

Data accessibility. The datasets supporting this article have been uploaded to Dryad Digital Repository: https://doi.org/10.5061/dryad.pk0p2ngm3 [37].

Authors' contributions. T.-Y.C. contributed in developing the idea, data analysis, execution and writing of the paper. T.K. contributed in the development of the walking robot, robotic feet, axial load machine and execution of experiments. N.O. provided professional advice on biomechanics, which helped in improving this manuscript. K.H. directed the project and contributed to the article conception, robot and foot developments, and the final manuscript. All the authors gave their final approval for publication.

Competing interests. We declare we have no competing interests.

Funding. This work was supported by JSPS KAKENHI 23220004.

Acknowledgements. The authors are grateful to Dr Xiangxiao Liu of Swiss Federal Institute of Technology in Lausanne (EPFL) for his suggestions/efforts on this research. The authors would also like to thank Mr Yusuke Seguchi of Osaka University for his help in the development of the axial load machine in this study, and Mr Arne Hitzmann, Ryu Takahashi and Tyler Kessler of Osaka University for their help during this study. Editage Ltd helped us in improving the English language quality of the manuscript.

# Appendix A

## A.1. Supplementary experimental data

The CoP positions and heel contact point (HCP) of each test foot of the axial loading experiment is listed in table 2. The HCP represents the consistent landmark of the foot. It is used to show the origin with respect to the CoP locations.

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
