## [Peer Review File · Royal Society Open Science]

Review History

RSOS-200323.R0 (Original submission)

Review form: Reviewer 1

Is the manuscript scientifically sound in its present form?

Yes

Are the interpretations and conclusions justified by the results?

No

Is the language acceptable?

Yes

Do you have any ethical concerns with this paper?

No

Have you any concerns about statistical analyses in this paper?

Yes

Recommendation?

Major revision is needed (please make suggestions in comments)

Comments to the Author(s)

Review: RSOS-200323

The authors investigated how the free moment changed with different robotic feet that either had a rigid foot, a transverse tarsal joint that was parallel to the ankle joint, and a transverse tarsal joint that rotated about an oblique axis. The authors demonstrate that integrating this oblique axis into a robotic foot increases the free moment during a standing-like situation and during bipedal walking, relative to the other feet tested. The authors do not explain the functional significance of the added free moment, only that it occurs in a higher magnitude with an oblique axis foot.

Major:

The authors indicate in the introduction that the free moment plays an important role in postural balance during human walking. However, the authors do not perform any analyses to show how integrating an oblique axis transverse tarsal joint contributes to postural control during walking. Further analysis is needed in this manuscript beyond the calculation of only the free moment. The authors postulate (in the discussion) that the increased free moment may help with balance. The authors have the data and thus should do more than postulate. For example, can the authors calculate the whole-body angular momentum with each of the different feet during walking?

Minor:

Remove spontaneous throughout the manuscript. The foot doesn't do anything spontaneous, the free moment is a reaction from forces applied to the foot structures.

Make sure to check to check the tense used throughout the paper to make sure it is consistent.

Abstract:

Page 3, Line 28: replace "the primary" with "a" as you only examined this one structure within the foot, not other structures.

Page 3, Lines 32-33: remove "We work on this hypothesis using a constructive approach" as it is not necessary.

Introduction:

The authors should reference that the transverse tarsal joint is also called the Chopart joint (which is used much more frequently in clinical research).

Page 3, Paragraphs 1 and 3: The authors frequently (8 times) use the following sentence structure: (author) observed/reported/built (etc.).... Please revise using concise language.

Page 3, Lines 10: remove "force" from "shock force absorption"

Page 3, Lines 19-21: "The ground reaction force and free moment both play important roles in postural balancing during biped walking [9,10]" Citation 10 (Willwacher et al., 2016) is not associated with walking and citation 9 (Herr and Popovic) is associated with angular momentum. The point the authors are trying to make is much more nuanced than this sentence is implying. Please revise.

Page 3, Line 24: change foots to foot's

Page 3, Line 27: do the author's mean standing postural control?

Page 3, Line 30: remove citation 13, they did not show that the foot's pronation/supination affects the free moment

Page 3, Lines 30-32: This seems like a large assumption that is not substantiated as there are multiple joints in the foot that contribute to pronation/supination.

Page 3, Lines 32-33: The following is not needed: "The transverse tarsal joint is a very complex and controversial structure" as this is shown in the next lines.

Page 3, Lines 45-47: Consider rephrasing the topic sentence to the following (or something similar): "Examining biomimetic robotic feet during standing and walking can reveal the role of different foot structures"

Page 4, Line 7: "build" to "built"

Page 4, Line 9: unnecessary comma

Section 2

Page 4, Lines 37-38: redundant sentence

Page 5, Line 11: use approximate rather than approach

Figure 2 caption: please expound upon your caption in order to tell readers where the transverse tarsal joint is located

Page 5, line 34: Should be Figure 3 rather than Figure 2

Page 5, line 34: add "of the" between "joint" and "robotic"

Page 6, line 11: "movement" to "moment"

Page 6, line 28: "generations of" to "generated by"

What surface did the authors use on the force plate? Changing the surface will change the friction and the free moment that is caused by the foot

Page 7: What was the walking speed of the robot?

Page 8, Table 1: In the text it states that there are no articulating toe joints but in the table, it shows that there are toe joints. Please update the text/table so that they are consistent.

What statistics were run by the authors? No statistical methods were described.

Results:

Figure 8: add a zero line

Discussion:

How do your free moment curves compare to human walking in magnitude (normalized to body weight and leg length) and in waveform?

Page 11, Lines 26-32: This is only a repetition/description of the results. Please move to results or remove all together.

Review form: Reviewer 2 (Ali Yawar)

Is the manuscript scientifically sound in its present form?

No

Are the interpretations and conclusions justified by the results?

No

Is the language acceptable?

No

Do you have any ethical concerns with this paper?

No

Have you any concerns about statistical analyses in this paper?

Yes

Recommendation?

Reject

Comments to the Author(s)

Summary of the manuscript:

This manuscript aims to use robotic feet to investigate the role of the transverse tarsal joint (TTJ) of the human foot in modulating vertical free moment at the foot-ground contact. To this end, the authors perform load testing on custom built robotic feet, following which they mount these feet on a bipedal walking robot. Three types of robotic feet are tested: one rigid foot with no internal degree of freedom, one foot with a transverse tarsal joint with axis parallel to the ankle axis, and one with the tarsal axis oblique based on published anatomical measurements in human feet. The feet with internal degrees of freedom also have pneumatic connectors to mimic the plantar fascia. Two experiments are reported. The first experiment involves vertical loading of each foot at the tibia and measurement of the resulting free vertical moment using a force plate under the foot. The foot with oblique TTJ produces a negative moment (about -0.5 Nm), while the rigid and non oblique feet produce small positive moments (about 0.3 Nm). In the second experiment, the same feet are mounted on a walking robot and the free moment is measured during stance as the robot walks on a force plate. The peak free moments produced during walking are lower for the foot with oblique TTJ than the other two. The main conclusion is that the results from the axial loading can be applied to walking. Therefore, under bodyweight, an oblique TTJ enables generation of free moments that can counter the moments generated by leg swinging, reducing the moment peaks.

Review:

The main conclusions in the manuscript are erroneous. Furthermore, the manuscript lacks necessary details for a reader to interpret and replicate the experiments. My main criticisms are elaborated below:

1. Incorrect conclusions:

In line 36-38 on pg. 10, the authors state, "According to the findings of the above axial loading experiment, the amplitude of the free moment can be reduced owing to the larger internal vertical moment generated via the oblique joint structure when body weight is applied".

Axially loading the foot with an oblique TTJ produced a negative free moment. The authors use this result to conclude that during walking with an oblique TTJ, bodyweight loading can generate a negative free moment and counter the positive moment produced by swing leg acceleration. This is incorrect, regardless of foot type. In the axial loading experiment the tibia is clamped to the loading machine. When axially loaded, rotation at the oblique tarsal joint causes the foot to deform. Friction between the foot and the force plate could then lead to a force couple and produce a free moment, which is ultimately supported by the tibial clamp. During walking bodyweight alone cannot generate any force couples under the foot. So, bodyweight cannot generate any free moment at the foot, and thus cannot counter free moments generated by swing leg acceleration (line 34).

2. Incomplete methods and data:

- a. Only one component of the moment is reported from the force plates, and so the overall loading picture is unknown.
- b. The repeatability the moment traces from the walking robot is unknown, since only the average trace is reported.
- b. Boundary conditions under the foot, and center of pressure location during load testing are not reported, which impede interpretation and replication.
- c. Basic experimental details are missing from the manuscript, for example: the robot's walking speed, justification for choice of plantar fascia stiffness, sampling rate of the force plate, details of filtering , software used for analysis etc.
- d. It is unclear to what extent results from the robot are applicable to human walking because no details of the robot are provided.
- e. The authors state p-values from the experiments without stating what statistical tests were performed.
- f. There is no description of summary statistics both in text and in bar charts.

Line by line comments (also marked up in the attached PDF):

P3, line 41: How can the reader interpret 500kPa? What type and size of actuator was used? Why was this value of the pressure selected?

P3, Figure 1: What is the boundary condition at the three contact points under the foot, and at the tibial attachment? Can the foot slide on the ground underneath, relative to each other? If not, how is the oblique foot different from the rigid foot? To what portion(s) of the stance phase is the axial loading picture applicable?

P4, line 39: What is "triplane motion"?

P5, line 9: Figures 1-3 show left feet so far. The direction of internal and external rotation are opposite for left and right feet and so this could be confusing to the reader.

P5, line 33: What were the boundary conditions under the foot?

P5, figure 4: The FM direction arrow is ambiguous. It would help for it to be placed on top of the Z axis, or with a different color

P6, line 7: Many details about this experiment are missing. At what speed did the robot walk? What determined the choice of muscle activations? Has this robot been previously characterized and published?

P6, line 13: Please summarize the robot's design here with appropriate citations if it was previously published.

P6, line 21: What empirical testing was performed and what was tuned?

P7, line 45: steady-state has not been defined

P7, line 46: p-values are from what statistical test?

P9, figure 9: What do the whiskers show? Why are they overlapping between conditions that are statistically significantly different?

P10, line 14: This is not required. In fact, having the cylinder piston rotation completely free would guarantee zero vertical moment on the foot.

P10, line 19: This sentence seems to imply that the slight rotation is significantly larger than the oblique

P10, line 20: The oblique joint does not spontaneously generate a "humanlike free moment". The moment is a consequence of the boundary conditions imposed on the foot. If the test was repeated with a 150N brick placed on top of the tibia, the vertical moment shown by the force plate would be zero.foot result.

P10, line 23: Bodyweight alone cannot generate vertical moments during walking.

P10, line 37: I am confused as to how body weight produces any vertical moment. Even if it somehow did, it would be a constant negative moment based on results from the axial loading, and would at best shift the entire moment trace and not reduce the peak to peak moment as is claimed.

P10, line 38: If the moment due to leg swing stays constant, shouldn't a smaller free moment lead to a larger yaw motion?

Review form: Reviewer 3

Is the manuscript scientifically sound in its present form?

Yes

Are the interpretations and conclusions justified by the results?

No

Is the language acceptable?

Yes

Do you have any ethical concerns with this paper?

No

Have you any concerns about statistical analyses in this paper?

No

Recommendation?

Major revision is needed (please make suggestions in comments)

Comments to the Author(s)

Thank you for giving me the opportunity to read the manuscript by Chen et al. This paper is quite compact and to-the-point which I like. I do not have many specific or detailed comments. Most of my concerns are on a general level, about the research questions posed, and how the data can help answer them. Some of my issues might be misunderstandings, in which case clarification would solve the issue.

This paper is generally well written in good scientific English. There are a few sections that need further language editing, most notably the Abstract, and some parts of the Introduction and Discussion.

Abstract

I am giving slightly more detailed comments on the Abstract because it is such an important part of the paper. I will refer to line numbers as printed in the margin.

Page 2 L22 "The" (not "a") human foot.

L23 "hypothesized": it has to be since the foot is the only mechanical link to the environment?

L30 "yaw" this might be personal preference but I feel this is an engineering term and "vertical moment" or "moment around the vertical axis" might be more suitable here.

L36, delete "An"

L43 "the robot" - must be introduced first.

L44 "constructive approach", I'm not sure whether this term (al in the title) will be understood by everyone.

One of the puzzling findings is that the results from the axial loading, and from the walking, are opposite in terms of free moment increase/reduction. The authors seem to favour the latter; yet when it comes to fundamental understanding one could argue that the former is more valuable as it isolates the function of the transverse tarsal joint. The robot walking would have much more confounding factors (mechanics higher up), and therefore the results would be harder to interpret unequivocally. For example (P7 L 20-21), the walking pattern was based on an existing one and then empirically tuned. It is not very clear in what way, and what the effect of the resulting walking pattern would be on the results.

Reading this manuscript made me wonder how much could we learn with simple modelling based on first principles?

There is a very relevant paper on the transverse tarsal arch that was published in Nature recently (probably after this manuscript was submitted) by Venkadesan et al. This paper should be referenced as it is very relevant and complementary to this one.

Page 3 (Introduction) L15-17: unclear sentence, please rephrase

On the whole, I think the Introduction (and to some extent the Abstract) should be clearer about why the vertical free moment is so important and why this study approach (including its research design, with the axial loading as well as robot gait) was decided upon.

It would be good to also mention the subtalar joint (which also has an oblique axis) and its function in gait, and also that what is called the "ankle joint" in this paper is effectively the talocrural joint which works much like a hinge joint with pure dorsi/ plantarflexion (e.g. P5 L37).

L54: Please explain the "constructive approach" in a bit more detail (what it is, and what its strengths and weaknesses are).

P4 L11: I just wanted to say I'm quite impressed by the technical skills of the team, and by their walking robot.

P6 L33: 150 N seems low if the idea is biomimicry.

P9 Figure 7: the values seem small to me, but the applied load (150 N) is also smaller than that seen during gait. I assume they scale linearly. If so, could you express the moment as a % to facilitate comparison with the literature?

P9, and Discussion. I am unsure how a rigid foot, or a foot with a non-oblique joint, could produce a free vertical moment. This is touched upon in the Discussion, but only for the axial loading where the authors suggest there was an issue with the connector. How can it be excluded that also in the walking experiment, mechanical issues confound the results? Especially since in Fig. 8 and further it can be seen that the moments are all relatively similar (at least, they don't change sign as in the axial loading test), so even small effect might affect the data.

In the Discussion (P11 L17 etc.) it is argued that the error due to the connection is small, but the data show it is approximately 50% of the moment measured for the oblique foot (with a different sign), which I would not consider small.

As a side note, Figure 8 is not referenced to in the text. In the figure legend, please explain how you defined mid-stance.

P11 (Discussion) L29-33: this section is not very clear to me, please rephrase.

P12 L6: Please rephrase the Ethics statement.

Decision letter (RSOS-200323.R0)

Dear Mr Chen:

Manuscript ID RSOS-200323 entitled "Free Moment induced by Oblique Transverse Tarsal Joint: Investigation by constructive approach" which you submitted to Royal Society Open Science, has been reviewed. The comments from reviewers are included at the bottom of this letter.

In view of the criticisms of the reviewers, the manuscript has been rejected in its current form. However, a new manuscript may be submitted which takes into consideration these comments.

Please note that resubmitting your manuscript does not guarantee eventual acceptance, and that your resubmission will be subject to peer review before a decision is made.

Your resubmitted manuscript should be submitted by 09-Nov-2020. If you are unable to submit by this date please contact the Editorial Office.

on behalf of Professor Madhusudhan Venkadesan (Associate Editor) and R. Kerry Rowe (Subject Editor)
openscience@royalsociety.org

Associate Editor Comments to Author (Professor Madhusudhan Venkadesan):

Associate Editor: 1

Comments to the Author:

The referees were appreciative of your clear writing but found substantial problems related to the scientific soundness of the result. They have also questioned the applicability of the result in the context of walking. I hope these reviews are helpful in improving the manuscript to clarify how and when the findings on the free negative moment of the foot may apply during walking. This is particularly important because the bodyweight load on the shank crucially differs from the clamp in the testing experiments in that the bodyweight cannot apply force couples. As a result, there may be no negative free moment due to an oblique axis of the transverse tarsal joint, which is a major point of the paper. In addition to this core criticism, the referees also found the paper lacking sufficient details to aid in scientific reproducibility.

I will be happy to reconsider the manuscript if the authors are able to address the issues of scientific correctness that have been raised by the reviews, in addition to the need for sufficient details on the methods and numerous other points that have been raised.

Associate Editor: 2

Comments to the Author:

(There are no comments.)

Reviewers' Comments to Author:

Reviewer: 1

Comments to the Author(s)

Review: RSOS-200323

The authors investigated how the free moment changed with different robotic feet that either had a rigid foot, a transverse tarsal joint that was parallel to the ankle joint, and a transverse tarsal joint that rotated about an oblique axis. The authors demonstrate that integrating this oblique axis into a robotic foot increases the free moment during a standing-like situation and during bipedal walking, relative to the other feet tested. The authors do not explain the functional significance of the added free moment, only that it occurs in a higher magnitude with an oblique axis foot.

Major:

The authors indicate in the introduction that the free moment plays an important role in postural balance during human walking. However, the authors do not perform any analyses to show how integrating an oblique axis transverse tarsal joint contributes to postural control during walking. Further analysis is needed in this manuscript beyond the calculation of only the free moment. The authors postulate (in the discussion) that the increased free moment may help with balance. The

authors have the data and thus should do more than postulate. For example, can the authors calculate the whole-body angular momentum with each of the different feet during walking?

Minor:

Remove spontaneous throughout the manuscript. The foot doesn't do anything spontaneous, the free moment is a reaction from forces applied to the foot structures.

Make sure to check to check the tense used throughout the paper to make sure it is consistent.

Abstract:

Page 3, Line 28: replace "the primary" with "a" as you only examined this one structure within the foot, not other structures.

Page 3, Lines 32-33: remove "We work on this hypothesis using a constructive approach" as it is not necessary.

Introduction:

The authors should reference that the transverse tarsal joint is also called the Chopart joint (which is used much more frequently in clinical research).

Page 3, Paragraphs 1 and 3: The authors frequently (8 times) use the following sentence structure: (author) observed/reported/built (etc.).... Please revise using concise language.

Page 3, Lines 10: remove "force" from "shock force absorption"

Page 3, Lines 19-21: "The ground reaction force and free moment both play important roles in postural balancing during biped walking [9,10]" Citation 10 (Willwacher et al., 2016) is not associated with walking and citation 9 (Herr and Popovic) is associated with angular momentum. The point the authors are trying to make is much more nuanced than this sentence is implying. Please revise.

Page 3, Line 24: change foots to foot's

Page 3, Line 27: do the author's mean standing postural control?

Page 3, Line 30: remove citation 13, they did not show that the foot's pronation/supination affects the free moment

Page 3, Lines 30-32: This seems like a large assumption that is not substantiated as there are multiple joints in the foot that contribute to pronation/supination.

Page 3, Lines 32-33: The following is not needed: "The transverse tarsal joint is a very complex and controversial structure" as this is shown in the next lines.

Page 3, Lines 45-47: Consider rephrasing the topic sentence to the following (or something similar): "Examining biomimetic robotic feet during standing and walking can reveal the role of different foot structures"

Page 4, Line 7: "build" to "built"

Page 4, Line 9: unnecessary comma

Section 2

Page 4, Lines 37-38: redundant sentence

Page 5, Line 11: use approximate rather than approach

Figure 2 caption: please expound upon your caption in order to tell readers where the transverse tarsal joint is located

Page 5, line 34: Should be Figure 3 rather than Figure 2

Page 5, line 34: add "of the" between "joint" and "robotic"

Page 6, line 11: "movement" to "moment"

Page 6, line 28: "generations of" to "generated by"

What surface did the authors use on the force plate? Changing the surface will change the friction and the free moment that is caused by the foot

Page 7: What was the walking speed of the robot?

Page 8, Table 1: In the text it states that there are no articulating toe joints but in the table, it shows that there are toe joints. Please update the text/table so that they are consistent.

What statistics were run by the authors? No statistical methods were described.

Results:

Figure 8: add a zero line

Discussion:

How do your free moment curves compare to human walking in magnitude (normalized to body weight and leg length) and in waveform?

Page 11, Lines 26-32: This is only a repetition/description of the results. Please move to results or remove all together.

Reviewer: 2

Comments to the Author(s)

Summary of the manuscript:

This manuscript aims to use robotic feet to investigate the role of the transverse tarsal joint (TTJ) of the human foot in modulating vertical free moment at the foot-ground contact. To this end, the authors perform load testing on custom built robotic feet, following which they mount these feet on a bipedal walking robot. Three types of robotic feet are tested: one rigid foot with no internal degree of freedom, one foot with a transverse tarsal joint with axis parallel to the ankle axis, and one with the tarsal axis oblique based on published anatomical measurements in human feet. The feet with internal degrees of freedom also have pneumatic connectors to mimic the plantar fascia. Two experiments are reported. The first experiment involves vertical loading of each foot at the tibia and measurement of the resulting free vertical moment using a force plate under the foot. The foot with oblique TTJ produces a negative moment (about -0.5 Nm), while the rigid and non oblique feet produce small positive moments (about 0.3 Nm). In the second experiment, the same feet are mounted on a walking robot and the free moment is measured during stance as the robot

walks on a force plate. The peak free moments produced during walking are lower for the foot with oblique TTJ than the other two. The main conclusion is that the results from the axial loading can be applied to walking. Therefore, under bodyweight, an oblique TTJ enables generation of free moments that can counter the moments generated by leg swinging, reducing the moment peaks.

Review:

The main conclusions in the manuscript are erroneous. Furthermore, the manuscript lacks necessary details for a reader to interpret and replicate the experiments. My main criticisms are elaborated below:

1. Incorrect conclusions:

In line 36-38 on pg. 10, the authors state, "According to the findings of the above axial loading experiment, the amplitude of the free moment can be reduced owing to the larger internal vertical moment generated via the oblique joint structure when body weight is applied".

Axially loading the foot with an oblique TTJ produced a negative free moment. The authors use this result to conclude that during walking with an oblique TTJ, bodyweight loading can generate a negative free moment and counter the positive moment produced by swing leg acceleration. This is incorrect, regardless of foot type. In the axial loading experiment the tibia is clamped to the loading machine. When axially loaded, rotation at the oblique tarsal joint causes the foot to deform. Friction between the foot and the force plate could then lead to a force couple and produce a free moment, which is ultimately supported by the tibial clamp. During walking bodyweight alone cannot generate any force couples under the foot. So, bodyweight cannot generate any free moment at the foot, and thus cannot counter free moments generated by swing leg acceleration (line 34).

2. Incomplete methods and data:

- a. Only one component of the moment is reported from the force plates, and so the overall loading picture is unknown.
- b. The repeatability of the moment traces from the walking robot is unknown, since only the average trace is reported.
- b. Boundary conditions under the foot, and center of pressure location during load testing are not reported, which impede interpretation and replication.
- c. Basic experimental details are missing from the manuscript, for example: the robot's walking speed, justification for choice of plantar fascia stiffness, sampling rate of the force plate, details of filtering, software used for analysis etc.
- d. It is unclear to what extent results from the robot are applicable to human walking because no details of the robot are provided.
- e. The authors state p-values from the experiments without stating what statistical tests were performed.
- f. There is no description of summary statistics both in text and in bar charts.

Line by line comments (also marked up in the attached PDF):

P3, line 41: How can the reader interpret 500kPa? What type and size of actuator was used? Why was this value of the pressure selected?

P3, Figure 1: What is the boundary condition at the three contact points under the foot, and at the tibial attachment? Can the foot slide on the ground underneath, relative to each other? If not, how is the oblique foot different from the rigid foot? To what portion(s) of the stance phase is the axial loading picture applicable?

P4, line 39: What is "triplane motion"?

P5, line 9: Figures 1-3 show left feet so far. The direction of internal and external rotation are opposite for left and right feet and so this could be confusing to the reader.

P5, line 33: What were the boundary conditions under the foot?

P5, figure 4: The FM direction arrow is ambiguous. It would help for it to be placed on top of the Z axis, or with a different color

P6, line 7: Many details about this experiment are missing. At what speed did the robot walk? What determined the choice of muscle activations? Has this robot been previously characterized and published?

P6, line 13: Please summarize the robot's design here with appropriate citations if it was previously published.

P6, line 21: What empirical testing was performed and what was tuned?

P7, line 45: steady-state has not been defined

P7, line 46: p-values are from what statistical test?

P9, figure 9: What do the whiskers show? Why are they overlapping between conditions that are statistically significantly different?

P10, line 14: This is not required. In fact, having the cylinder piston rotation completely free would guarantee zero vertical moment on the foot.

P10, line 19: This sentence seems to imply that the slight rotation is significantly larger than the oblique

P10, line 20: The oblique joint does not spontaneously generate a "humanlike free moment". The moment is a consequence of the boundary conditions imposed on the foot. If the test was repeated with a 150N brick placed on top of the tibia, the vertical moment shown by the force plate would be zero.foot result.

P10, line 23: Bodyweight alone cannot generate vertical moments during walking.

P10, line 37: I am confused as to how body weight produces any vertical moment. Even if it somehow did, it would be a constant negative moment based on results from the axial loading, and would at best shift the entire moment trace and not reduce the peak to peak moment as is claimed.

P10, line 38: If the moment due to leg swing stays constant, shouldn't a smaller free moment lead to a larger yaw motion?

Reviewer: 3

Comments to the Author(s)

Thank you for giving me the opportunity to read the manuscript by Chen et al. This paper is quite compact and to-the-point which I like. I do not have many specific or detailed comments. Most of my concerns are on a general level, about the research questions posed, and how the data

can help answer them. Some of my issues might be misunderstandings, in which case clarification would solve the issue.

This paper is generally well written in good scientific English. There are a few sections that need further language editing, most notably the Abstract, and some parts of the Introduction and Discussion.

Abstract

I am giving slightly more detailed comments on the Abstract because it is such an important part of the paper. I will refer to line numbers as printed in the margin.

Page 2 L22 “The” (not “a”) human foot.

L23 “hypothesized”: it has to be since the foot is the only mechanical link to the environment?

L30 “yaw” this might be personal preference but I feel this is an engineering term and “vertical moment” or “moment around the vertical axis” might be more suitable here.

L36, delete “An”

L43 “the robot” – must be introduced first.

L44 “constructive approach”, I’m not sure whether this term (al in the title) will be understood by everyone.

One of the puzzling findings is that the results from the axial loading, and from the walking, are opposite in terms of free moment increase/reduction. The authors seem to favour the latter; yet when it comes to fundamental understanding one could argue that the former is more valuable as it isolates the function of the transverse tarsal joint. The robot walking would have much more confounding factors (mechanics higher up), and therefore the results would be harder to interpret unequivocally. For example (P7 L 20-21), the walking pattern was based on an existing one and then empirically tuned. It is not very clear in what way, and what the effect of the resulting walking pattern would be on the results.

Reading this manuscript made me wonder how much could we learn with simple modelling based on first principles?

There is a very relevant paper on the transverse tarsal arch that was published in Nature recently (probably after this manuscript was submitted) by Venkadesan et al. This paper should be referenced as it is very relevant and complementary to this one.

Page 3 (Introduction) L15-17: unclear sentence, please rephrase

On the whole, I think the Introduction (and to some extent the Abstract) should be clearer about why the vertical free moment is so important and why this study approach (including its research design, with the axial loading as well as robot gait) was decided upon.

It would be good to also mention the subtalar joint (which also has an oblique axis) and its function in gait, and also that what is called the “ankle joint” in this paper is effectively the talocrural joint which works much like a hinge joint with pure dorsi/ plantarflexion (e.g. P5 L37).

L54: Please explain the “constructive approach” in a bit more detail (what it is, and what its strengths and weaknesses are).

P4 L11: I just wanted to say I’m quite impressed by the technical skills of the team, and by their walking robot.

P6 L33: 150 N seems low if the idea is biomimicry.

P9 Figure 7: the values seem small to me, but the applied load (150 N) is also smaller than that seen during gait. I assume they scale linearly. If so, could you express the moment as a % to facilitate comparison with the literature?

P9, and Discussion. I am unsure how a rigid foot, or a foot with a non-oblique joint, could produce a free vertical moment. This is touched upon in the Discussion, but only for the axial loading where the authors suggest there was an issue with the connector. How can it be excluded that also in the walking experiment, mechanical issues confound the results? Especially since in Fig. 8 and further it can be seen that the moments are all relatively similar (at least, they don't change sign as in the axial loading test), so even small effect might affect the data.

In the Discussion (P11 L17 etc.) it is argued that the error due to the connection is small, but the data show it is approximately 50% of the moment measured for the oblique foot (with a different sign), which I would not consider small.

As a side note, Figure 8 is not referenced to in the text. In the figure legend, please explain how you defined mid-stance.

P11 (Discussion) L29-33: this section is not very clear to me, please rephrase.

P12 L6: Please rephrase the Ethics statement.

Author's Response to Decision Letter for (RSOS-200323.R0)

See Appendix A.

RSOS-201947.R0

Review form: Reviewer 1

Is the manuscript scientifically sound in its present form?

Yes

Are the interpretations and conclusions justified by the results?

Yes

Is the language acceptable?

Yes

Do you have any ethical concerns with this paper?

No

Have you any concerns about statistical analyses in this paper?

No

Recommendation?

Accept with minor revision (please list in comments)

Comments to the Author(s)

I appreciate the time and effort that the authors have put into updating their manuscript. There are several minor suggestions to enhance this work.

Thank you for including more details regarding your experimental set up.

I would encourage the authors to have an overall statistical analysis portion after section “e” of their methods so as to combine and provide the analysis performed to the readers in one section. I found myself going back and forth to figure out what statistical analyses were performed for the walking experiment (these were not explicitly stated, it is assumed that they are the same as load experiment).

I am curious about the vertical GRFs experienced during the walking portion of the experiment. How does it compare to the 150N placed on the foot during the static loading portion of the experiment? It is interesting to me because the loading experiment had an order of magnitude smaller free moment than the walking experiment.

Please include a limitation that the walking experiment was not performed at steady-state. Though it is noted the first step was very repeatable between trials. In my experience, steady-state walking (in humans and models) would be indicative of walking indefinitely (or almost indefinitely).

I also did enjoy seeing your robot walking (in the supplementary material).

Please check all grammar.

Review form: Reviewer 2 (Ali Yawar)

Is the manuscript scientifically sound in its present form?

No

Are the interpretations and conclusions justified by the results?

No

Is the language acceptable?

No

Do you have any ethical concerns with this paper?

No

Have you any concerns about statistical analyses in this paper?

No

Recommendation?

Major revision is needed (please make suggestions in comments)

Comments to the Author(s)

I appreciate the effort the authors have put towards addressing my criticisms. Most of the technical details necessary to replicate their experiments are now reported in the revised manuscript. When considered independently, both the reported experiments (static load testing of the feet, and ground reaction measurement in the walking robot) are technically sound.

However, the major scientific flaw of the original submission still remains in this revised version: the results of the static loading experiment cannot be applied to the walking experiment because the boundary conditions at the tibia are fundamentally different. The connection between these two experiments is the central message of the manuscript, so I cannot recommend it for publication in its current form. The design, characterization, and pneumatic control of the foot and the walking robot are important in their own right and would make valuable contributions to the literature once appropriately rewritten to remove incorrect inferences.

Here is a summary of the scientific flaw as I see it:

In the static loading experiment, as the oblique foot rotates externally under axial loading, the clamped tibia provides the reaction moment to maintain static equilibrium, which is measured by the force plate under the foot. This reaction moment appears because the tibial clamp resists rotation and thus can apply force couples. On the contrary, during the single support phase of walking or running, the boundary condition at the tibia is free. The bodyweight load alone cannot generate any force couples or free moments. Thus, the authors' claim that the oblique foot generates a countering moment during walking and reduces the peaks in the measured free moment is incorrect.

Alternative explanation of the authors' measurement:

The authors report a reduction in the peaks of the free moment with the oblique feet compared to rigid/non-oblique feet in walking. Here is a possible alternative explanation of what could be happening: In response to contralateral leg swing, the free-moment transients at the supporting foot depend on how mobile the foot/ankle is about the vertical axis. When the supporting foot/ankle is rigid, and the body cannot spin about the vertical axis, the measured free moment under the foot would be the highest. On the other hand, if the ankle was an ideal ball and socket joint, the measured free moment would be zero, and the body would just spin in response to leg swing. Since the oblique foot has a degree of freedom about the vertical axis, it could be that the free moment transients associated with contralateral leg swing are smaller than with the rigid foot.

Some technical details that are necessary for replication and interpretation are still missing:

1. Fig 10, x-axis label: It is unclear what "Load" refers to.
2. In Table 2, the origin with respect to which the CoP location is measured is not reported. The authors should report the location of the center of pressure with respect to a consistent landmark on the foot (say, the heel contact point) so that the relative location of the CoP can be interpreted between the different feet.

Decision letter (RSOS-201947.R0)

Dear Mr Chen

The Editors assigned to your paper RSOS-201947 "Free Moment induced by Oblique Transverse Tarsal Joint: Investigation by constructive approach" have now received comments from reviewers and would like you to revise the paper in accordance with the reviewer comments and any comments from the Editors. Please note this decision does not guarantee eventual acceptance.

Please submit your revised manuscript and required files (see below) no later than 21 days from today's (ie 09-Feb-2021) date. Note: the ScholarOne system will 'lock' if submission of the revision is attempted 21 or more days after the deadline. If you do not think you will be able to meet this deadline please contact the editorial office immediately.

on behalf of Professor Madhusudhan Venkadesan (Associate Editor) and R. Kerry Rowe (Subject Editor)
openscience@royalsociety.org

Reviewer comments to Author:
Reviewer: 1

Comments to the Author(s)

I appreciate the time and effort that the authors have put into updating their manuscript. There are several minor suggestions to enhance this work.

Thank you for including more details regarding your experimental set up.

I would encourage the authors to have an overall statistical analysis portion after section "e" of their methods so as to combine and provide the analysis performed to the readers in one section. I found myself going back and forth to figure out what statistical analyses were performed for the walking experiment (these were not explicitly stated, it is assumed that they are the same as load experiment).

I am curious about the vertical GRFs experienced during the walking portion of the experiment. How does it compare to the 150N placed on the foot during the static loading portion of the

experiment? It is interesting to me because the loading experiment had an order of magnitude smaller free moment than the walking experiment.

Please include a limitation that the walking experiment was not performed at steady-state. Though it is noted the first step was very repeatable between trials. In my experience, steady-state walking (in humans and models) would be indicative of walking indefinitely (or almost indefinitely).

I also did enjoy seeing your robot walking (in the supplementary material).

Please check all grammar.

Reviewer: 2

Comments to the Author(s)

I appreciate the effort the authors have put towards addressing my criticisms. Most of the technical details necessary to replicate their experiments are now reported in the revised manuscript. When considered independently, both the reported experiments (static load testing of the feet, and ground reaction measurement in the walking robot) are technically sound. However, the major scientific flaw of the original submission still remains in this revised version: the results of the static loading experiment cannot be applied to the walking experiment because the boundary conditions at the tibia are fundamentally different. The connection between these two experiments is the central message of the manuscript, so I cannot recommend it for publication in its current form. The design, characterization, and pneumatic control of the foot and the walking robot are important in their own right and would make valuable contributions to the literature once appropriately rewritten to remove incorrect inferences.

Here is a summary of the scientific flaw as I see it:

In the static loading experiment, as the oblique foot rotates externally under axial loading, the clamped tibia provides the reaction moment to maintain static equilibrium, which is measured by the force plate under the foot. This reaction moment appears because the tibial clamp resists rotation and thus can apply force couples. On the contrary, during the single support phase of walking or running, the boundary condition at the tibia is free. The bodyweight load alone cannot generate any force couples or free moments. Thus, the authors' claim that the oblique foot generates a countering moment during walking and reduces the peaks in the measured free moment is incorrect.

Alternative explanation of the authors' measurement:

The authors report a reduction in the peaks of the free moment with the oblique feet compared to rigid/non-oblique feet in walking. Here is a possible alternative explanation of what could be happening: In response to contralateral leg swing, the free-moment transients at the supporting foot depend on how mobile the foot/ankle is about the vertical axis. When the supporting foot/ankle is rigid, and the body cannot spin about the vertical axis, the measured free moment under the foot would be the highest. On the other hand, if the ankle was an ideal ball and socket joint, the measured free moment would be zero, and the body would just spin in response to leg swing. Since the oblique foot has a degree of freedom about the vertical axis, it could be that the free moment transients associated with contralateral leg swing are smaller than with the rigid foot.

Some technical details that are necessary for replication and interpretation are still missing:

1. Fig 10, x-axis label: It is unclear what "Load" refers to.
2. In Table 2, the origin with respect to which the CoP location is measured is not reported. The authors should report the location of the center of pressure with respect to a consistent landmark

on the foot (say, the heel contact point) so that the relative location of the CoP can be interpreted between the different feet.

===PREPARING YOUR MANUSCRIPT===

===PREPARING YOUR REVISION IN SCHOLARONE===

- 1) One version identifying all the changes that have been made (for instance, in coloured highlight, in bold text, or tracked changes);
 - 2) A 'clean' version of the new manuscript that incorporates the changes made, but does not highlight them.
 - An individual file of each figure (EPS or print-quality PDF preferred [either format should be produced directly from original creation package], or original software format).
 - An editable file of each table (.doc, .docx, .xls, .xlsx, or .csv).
 - An editable file of all figure and table captions.
- Note: you may upload the figure, table, and caption files in a single Zip folder.
- Any electronic supplementary material (ESM).
 - If you are requesting a discretionary waiver for the article processing charge, the waiver form must be included at this step.
 - If you are providing image files for potential cover images, please upload these at this step, and inform the editorial office you have done so. You must hold the copyright to any image provided.
 - A copy of your point-by-point response to referees and Editors. This will expedite the preparation of your proof.

- Ensure that your data access statement meets the requirements at <https://royalsociety.org/journals/authors/author-guidelines/#data>. You should ensure that you cite the dataset in your reference list. If you have deposited data etc in the Dryad repository, please include both the 'For publication' link and 'For review' link at this stage.
- If you are requesting an article processing charge waiver, you must select the relevant waiver option (if requesting a discretionary waiver, the form should have been uploaded at Step 3 'File upload' above).
- If you have uploaded ESM files, please ensure you follow the guidance at <https://royalsociety.org/journals/authors/author-guidelines/#supplementary-material> to include a suitable title and informative caption. An example of appropriate titling and captioning may be found at https://figshare.com/articles/Table_S2_from_Is_there_a_trade-off_between_peak_performance_and_performance_breadth_across_temperatures_for_aerobic_sc_ope_in_teleost_fishes_/3843624.

Author's Response to Decision Letter for (RSOS-201947.R0)

See Appendix B.

Decision letter (RSOS-201947.R1)

Dear Mr Chen,

It is a pleasure to accept your manuscript entitled "Free Moment induced by Oblique Transverse Tarsal Joint: Investigation by constructive approach" in its current form for publication in Royal Society Open Science. The comments of the reviewer(s) who reviewed your manuscript are included at the foot of this letter.

Please ensure:

1. That you include your correct, active Dryad link for publication in the proofing stage;
2. That you provide an updated email address for kawakami.takahiko@arl.sys.es.osaka-u.ac.jp - Dr Kawakami is not currently receiving emails from our addresses.

on behalf of Professor Madhusudhan Venkadesan (Associate Editor) and R. Kerry Rowe (Subject Editor)
openscience@royalsociety.org

Associate Editor Comments to Author (Professor Madhusudhan Venkadesan):

Associate Editor

Comments to the Author:

The authors have responded adequately to the previous reviews, including acknowledging the limitations that were raised by both the referees.

Read Royal Society Publishing's blog:
<https://royalsociety.org/blog/blogsearchpage/?category=Publishing>

Appendix A

Responses

Response to Reviewer 1

Thanks for your kind comments and very detailed advice on the draft.

(1) About the major comment:

The issue of the postural control evaluation:

About this description, we just want to highlight the importance of the free moment investigation in the referenced article. We apologize for the misunderstanding.

We only focus on the influence of the free moment that is brought from the oblique transverse tarsal joint in this study, so we did not conduct a further experiment about this issue.

Meanwhile, due to the technical difficulty and resource limitation, it is difficult to measure reliable data for evaluation at the current state.

Therefore, we decided to remove the description of the postural control in the introduction.

We hope you understand our difficulties, and that we can solve this difficulty and conduct a more detailed study about this issue in the near future.

(2) About the minor comments:

The grammar and table mistakes have been updated and revised.

(1) **What surface did the authors use on the force plate?**

We put a foam mat composed of urethane.

(The static and kinetic friction coefficients are approximately 0.3 and 0.18)

(2) **Discussion:**

How do your free moment curves compare to human walking in magnitude (normalized to body weight and leg length) and in waveform?

The figure below is our walking results after the correction of the direction normalized with the robot's body weight and leg length.

Comparing to the human's waveform, we found that shape of the oblique foot's waveform is the most similar to the human's results[6,10].

The magnitude was smaller than an order.

(a) rigid

(b) non-oblique

(c) oblique

Response to Reviewer 2

To respond clearly, we answer the questions by following the page.

[1] Author-supplied statements page:

Data:

Thanks for your reminder. It has been changed to “Please contact the authors for accessing the detailed design and experimental data.

P3: line 41: How can the reader interpret 500kPa?

It can be interpreted as an elastic plantar fascia.

What type and size of actuator was used?

Why was this value of the pressure selected?

We use the McKibben type pneumatic muscle.

The inner tube diameter is 7 mm

The tube length is 120 mm

The stiffness of the muscle is approximately 9.8 N/mm with 500kPa.

Due to this, the stiffness is very close to the investigation of the human elastic plantar fascia [31].

We also refer to the elastic plantar fascia design parameters of other robotic studies [17, 32].

The pressurized parameter range is from 250kPa to 550kPa [17] and the stiffness is 12.5 N/mm [32].

Therefore, we decided to select this pressurized parameter.

(a) Pressurized status (b) Exhaust status

P3: Figure 1: What is the boundary condition at the three contact points under the foot, and the tibial attachment?

Can the foot slide on the ground underneath, relative to each other?

If not, how is the oblique foot different from the rigid foot?

In this study, the plantar surface boundary condition was sliding.

To what portion (s) of the stance phase is the axial loading picture applicable?

We think the results can be applied in the period of the latter half of the stance phase (i.e. the duration approximately 38% to 78% of the stance phase).

According to the walking experimental results, the generated external free moment might be reduced by the induced internal vertical free moment of the oblique foot.

P4: line 39: What is "triplane motion"?

We apologize for the typographical error.

It has been corrected to tri-planar motion.

P4: line 42: How is the stiffness controlled?

The stiffness of the McKibben type pneumatic muscle is able to be controlled by controlling the air pressure [30].

P5, line 9 Figure 1-3 show left feet so far. The direction of internal and external rotation is opposite for left and right feet and so this could be confusing to the reader.

Thank you for the advice, we used the same direction in all the pictures in the new draft.

P5, figure 4: The FM direction arrow is ambiguous. It would help for it to be placed on top of the Z axis, or with a different color

Thank you for mentioning this. We revised the arrow for clear indication.

P5:line 33: What were the boundary conditions under the foot?

The boundary condition was sliding.

*P6: line 7 Many details about this experiment are missing.
At what speed did the robot walk?*

The walking speed was 0.5 m/s

What determined the choice of muscle activations?

To make the musculoskeletal robot realize the flexion/extension of hip, knee, and ankle joints and to produce human-like walking.

We integrate the reports of the human muscles' activations during walking[36], muscles' antagonist motion, and the studies of the musculoskeletal robot [17,28] to determine the choice of muscle activations.

Has this robot been previously characterized and published?

P6:line 13 Please summarize the robot's design here with appropriate citations if was previously published.

It has not been published before.

This is a new robot that is modified from our previous studies[17][28].

P6: line 22 What empirical testing was performed and what was tuned?

We tuned the timing of the activation of the pneumatic solenoid valves to drive the muscles.

The timing was determined by empirical testing to make sure the robot can walk over 3 steps without falling when equipped with the tested feet.

A single leg’s control pattern is shown below.

At the heel of the left foot was installed a load cell (Tec Gihan Co., Ltd USL06-H5-500N) to sense heel contact.

For example, when the left leg senses the contact signal then the stand phase control pattern will be activated. At the same time, the right leg will be controlled by the swing phase control pattern. With this interaction, the robot can walk continuously.

One leg sequence control pattern

P7: line46 steady-state has not been defined.

We apologize that we did not explain this clearly.

The steady-state means that the three contact points of the calcaneus, first metatarsal, and fifth metatarsal are all in contact with the ground.

However, we think this description might be misleading to the reader, then we decided to not include this description in the new draft.

P9: Walking Results:

What do the whiskers show?

Why are they overlapping between conditions that are statistically significantly different?

About this statistical significance, it was mislabeled.

In the new draft, we corrected the figure's direction and filter.

Therefore, the values of the walking experimental results have been changed.

Also, we follow the other reviewers' advice, and the results were normalized by the body weight and leg length of the walking robot.

The new whiskers chart and bar chart are shown below.

In the new chart, it has no extreme values.

The whiskers chart of the walking experimental results

Numerous related articles used bar charts to show the results.

To clearly show the quantification data, we believe that, with appropriate statistical analysis, the bar chart is enough to show the result.

P10: The incorrect conclusion:

Thank you for reminding us of the incorrect conclusion.

We also apologize that we did not explain very clearly.

In the new draft, the conclusion of the axial load experiments has been revised to:

“By observing the movement of the oblique foot, the forefoot tends to externally rotate while bearing the axial loading.

Therefore, this movement may induce friction to generate the resisting internal free moment. ”

About the conclusion of the walking experiment, we find that the coordinate of the walking results of the previous draft was opposite to our definition. We apologize for the confusion with the previous results. The detailed revised content is added in the new draft.

Here, we simply brief the conclusion of the walking experiment.

Comparing the waveforms of the tested feet, the change of the oblique foot during the latter half (i.e. the duration approximately 38% to 78% of the stance phase) was the

smoothest in comparison to the rigid and non-oblique feet.

The smallest positive peak occurred with the oblique foot condition.

It also had statistical significance between the rigid (p-value < 0.01) and non-oblique (p-value < 0.05) conditions.

Applying the finding of the axial experiment, the walking results suggest that the external free moment generated during the latter half of the stance phase was reduced by the internal free moment induced by the oblique foot.

These results suggest that the foot mechanism with the oblique transverse tarsal joint might affect the transverse motion of bipedal walking and highlights the importance of the morphology of the human foot.

The repeatability of the moment traces from the walking robot is unknown, since only the average trace is reported.

In the new draft, we corrected the coordinates of the walking experiment, added the confidence interval, and processed by a custom low-pass filter.

The results were normalized by weight and leg length of the biped walking robot.

The updated figure is shown below.

The center of pressure location during load testing are not reported, which impede interpretation and replication.

The center of pressure (CoP) locations of the tested feet are listed below.

We added this table in the appendix section.

Type/ CoP	X (mm)	Y (mm)
Non-oblique foot :	-42.4 ± 0.1	12.4 ± 0.8
Oblique foot :	-20.6 ± 0.1	-7.1 ± 0.5
Rigid foot :	-35.8 ± 0.2	-7.9 ± 2.4

Basic experimental details are missing from the manuscript:

Thank you for these important suggestions.

The sampling rate of the force plate was 1KHz.

We used the median filter in the previous draft.

However, we found that numerous related articles use the lowpass filter.

To improve accuracy, we integrated python, Pandas, and Scipy when programming the filter for the new draft.

The filter is a 4th order low pass Butterworth filter with a 15Hz cut off frequency.

The authors state p-values from the experiments without stating what statistical tests were performed.

There is no description of summary statistics both in text and in bar charts.

In the previous draft, we only used the student T-Test for statistical analysis.

To improve accuracy, we decided to use the one-way ANOVA with Tukey's post honestly significant difference test in the new draft.

Response to Reviewer 3

One of the puzzling findings is that the results from the axial loading, and from the walking, are opposite in terms of free moment increase/reduction.

In the walking result, we apologize that we did not appropriately process the result.

After the recheck of the experimental data, we found the coordinates of the walking results which were shown in the previous draft were opposite to our definition.

In the new draft, we corrected the coordinates of the figure and supplemented the description for the unclear parts.

The authors seem to favor the latter; yet when it comes to fundamental understanding one could argue that the former is more valuable as it isolates the function of the transverse tarsal joint.

The robot walking would have more confounding factors (mechanics higher up), and therefore the results would be harder to interpret unequivocally.

For example (P7 L 20-21), the walking pattern was based on an existing one and then empirically tuned. It is not very clear in what way, and what the effect of the resulting walking pattern would be on the results.

We agree with your opinion.

However, we not only want to investigate the contribution of the oblique TTJ on the free moment, but also want to confirm whether or not this contribution or tendency occurs under the different upper boundary (i.e. free rotation of the tibia) condition.

Therefore, we think if we provide enough information on the walking experiment, there is a reason to conduct the walking experiment.

In the new draft, we not only provide information that was already shown in the previous version, but also added detailed information on the robot's design, experimental procedure, and sequence control chart of the muscle.

Especially with the robot's design, to clearly observe the contributions of the test feet, we try our best to remove the factor that may affect the free moment generation.

Therefore, the hip and knee joints of the walking robot are designed to only allow for flexion/extension and have no arm.

Although it can not perfectly reduce the mechanical influence, we think this design still can reduce the mechanical influence to some extent.

Reading this manuscript made me wonder how much could we learn with simple modeling based on first principles?

The human foot structure characteristics evolve from interacting with nature.

However, why these structures form these shapes is still not fully understood.

We think the simple model can let us understand how specific aspects deal with various environments step by step. It is the reason that we used the constructive approach for this study.

In addition, we can integrate other desired conditions to study the combined effects.

So, we think it still has many opportunities to develop.

We think we can discuss more on this issue.

There is a very relevant paper on the transverse tarsal arch that was published in Nature recently (probably after this manuscript was submitted) by Venkadesan et al. This paper should be referenced as it is very relevant and complementary to this one.

Thank you, it is good to know this helpful information.

They also constructed a simple structure to express the transverse tarsal joint's stiffness.

It would be good to also mention the subtalar joint (which also has an oblique axis) and its function in gait, and also that what is called the "ankle joint" in this paper is effectively the talocrural joint which works much like a hinge joint with pure dorsi/ plantarflexion (e.g. P5 L37).

Thank you for mentioning the subtalar joint.

We are running that research currently.

As you said, due to it also being an oblique axis, we think its deformation might also affect the generation of the free moment during the period from initial contact to mid-stance.

We think it is good advice to more clearly describe the joint that we designed.

In the new draft, we added the supplemental description for this joint.

L54: Please explain the “constructive approach” in a bit more detail (what it is, and what its strengths and weaknesses are).

The constructive approach is developed based on the embedded intelligence which is proposed by Prof. Rolf Pfeifer [26].

It is a research method that realizes a specific aspect or thing to discuss its contribution or influence on the real environment.

It can bring us to understand a complex system by starting with a simple model and adding to it step by step.

We also can integrate other aspects and control the boundary condition to design the desired observation size.

However, with this method, it is difficult to make a perfect system like the real world.

It only can reflect the real system to a certain extent.

Therefore, to understand the real-world system, we still need to combine it with conventional methods.

P4 L11: I just wanted to say I’m quite impressed by the technical skills of the team, and by their walking robot.

We appreciate that you like our work.

P6 L33: 150 N seems low if the idea is biomimicry.

We had three reasons for choosing this parameter for the experiments.

- (1) The loading limitation of the design of the oblique foot.
- (2) This parameter is close to our biped robot’s weight.
- (3) We want to compare the results of the cadaver foot.

P9 Figure 7: the values seem small to me, but the applied load (150 N) is also smaller than that seen during gait. I assume they scale linearly.

If so, could you express the moment as a % to facilitate comparison with the literature?

Thanks to this helpful advice, in the new draft, we express the axial loading experimental results by normalizing the applied load.

The below results were processed by calibrating the force plate with our ground setting.

With the one-way ANOVA with Tukey’s post HSD statistical analysis, all of the data sets have a significant difference between each other (p-value <0.01).

P9, and Discussion. I am unsure how a rigid foot, or a foot with a non-oblique joint, could produce a free vertical moment.

This is touched upon in the Discussion, but only for the axial loading where the authors suggest there was an issue with the connector.

How can it be excluded that also in the walking experiment, mechanical issues confound the results? Especially since in Fig. 8 and further it can be seen that the moments are all relatively similar (at least, they don't change sign as in the axial loading test), so even small effect might affect the data.

In the Discussion (P11 L17 etc.) it is argued that the error due to the connection is small, but the data show it is approximately 50% of the moment measured for the oblique foot (with a different sign), which I would not consider small.

After the investigation, we found that the reason that generated this puzzling result was that we forgot to calibrate the results with our ground setting (i.e. the green foam mat).

In the new draft, we showed all the results after calibration.

As a side note, Figure 8 is not referenced to in the text.

Thank you for mention it.

In the figure legend, please explain how you defined mid-stance.

While there are many definitions, “the point in walking at which the raised leg passes the grounded leg that is supporting the body's weight” is the most appropriate for this study.

At the same time, after rechecking the results, we found that the mid-stance phase of each foot was not the same, so we separately express the results in the new draft.

P11 (Discussion) L29-33: this section is not very clear to me, please rephrase.

P12 L6: Please rephrase the Ethics statement.

Thanks for the reminder, in the new draft the context has been corrected and revised.

Appendix B

Response

We greatly appreciate that all the Editors and reviewers took the time to give us kind and important advice on our submission in this difficult time.

We hope you are keeping well during this difficult and unusual time

In the new revision, the change is shown below.

1. Adding the limitation of the walking experiment
2. Adding a new section for the introduction of the Statistical method and Data Analysis
3. Changing of the ambiguous label "LOAD" to "AL" (Axial Load)
4. Changing the explanation of the walking experiment
5. Adding the description of the heel contact point
6. Revising the mislabeling of Table 2
7. Adding the heel contact points information into Table 2.
8. Adding the video of the left oblique foot's movement for the supplementary material.

Response to Reviewer 1

We appreciate your encouragement

- Comment 1:

I would encourage the authors to have an overall statistical analysis portion after section “e” of their methods so as to combine and provide the analysis performed to the readers in one section. I found myself going back and forth to figure out what statistical analyses were performed for the walking experiment (these were not explicitly stated, it is assumed that they are the same as load experiment).

Thank you for giving us advice.

We think it helps the reader quickly understand the statistical analyses that we used in all experiments.

In the revised draft, we added a new section for the introduction of the data analysis.

- Comment 2:

I am curious about the vertical GRFs experienced during the walking portion of the experiment. How does it compare to the 150N placed on the foot during the static loading portion of the experiment? It is interesting to me because the loading experiment had an order of magnitude smaller free moment than the walking experiment.

Thank you for mentioning the difference in the order of magnitude between these two experimental results.

We think the reason that causes this difference is the parameters of normalization.

In the vertical load experiment, following the previous advice of one of the reviewers, we only used the loading weight (150N) as the normalization parameter.

In the walking experiment, we used the weight of the walking robot (8.3 kg) and its leg length (0.65m) as the normalization parameters (i.e. the normalization parameter was approximately 52.9).

Figure.1 The measurement of the ground reaction force (GRF) of the walking experiment.

In this study, we only focus on the contribution of the structural influence of the free moment. However, we believe that the discussion of the ground reaction force may improve this study and helps us find a deeper contribution of the human oblique transverse tarsal joint. We think only to show the result of the ground reaction force without any discussion may mislead the reader.

So, we decided to show the above figure only in this response.

The relationship between the GRF and free moment is very interesting to us.

The details of the conversion between GRF and free moment by the foot's mechanism will be studied in the near future.

- Comment 3:

Please include a limitation that the walking experiment was not performed at steady-state. Though it is noted the first step was very repeatable between trials. In my experience, steady-state walking (in humans and models) would be indicative of walking indefinitely (or almost indefinitely).

Thanks for mentioning this important information.

We added the limitation in the section of the walking experiment.

Response to Reviewer 2

We appreciate your important advice and the kind alternative explanation
Also, we apologize for the incorrect explanation of the walking experiment.

- Comment 1:

About the scientific flaw of the walking experiment:

We appreciate that you gave us a very important comment.

We have carefully considered input.

We hope we do not misunderstand your major comment.

First, we agree with your opinion of the incorrect explanations of the experimental results.

As you say, due to the difference in the tibial condition of these two experiments (i.e. free and non-free), it can not directly apply to describe the free moment reduction in the walking experiment.

We apologize for our incorrect explanation of the finding of the walking experiment.

We also agree that your opinion of the contrary vertical moment and free moment can not be generated with bodyweight alone.

We believe the connection between the axial loading and walking experiments still needs a more detailed study to find the relationship.

However, after searching the articles and a great deal of discussion, we cannot find an appropriate explanation and theory to connect these two findings.

So, in this study, we think to explain the connection without appropriate evidence is very incorrect and careless.

Your alternative explanation inspired us to think about the ankle's movement of the oblique foot.

We think this movement could explain the reduction of the free moment of the walking experiment.

By observing the movement of the ankle under load on the oblique foot in the free tibia condition, we surprisingly found that the ankle's movement has an internal rotation element in the transverse plane.

It seems this transient internal rotation may generate a torque to resist the contrary moment being generated by the swing leg.

Unloaded

Under load

Figure 2. The movement of the left oblique foot (front view)

However, this hypothesis still needs more detailed research to confirm.

We are very sorry that we cannot confirm this at the current stage.

Besides, so the reader can easily understand this explanation, we also made a video of the oblique foot movement and uploaded it as supplementary material to Dryad.

The free moment reduction explanation of the revised draft is shown below.

By observing the movement of the oblique foot while bearing the pressing force (the left foot movement is shown in the supplementary video), we found that the ankle part of the oblique foot has partial internal rotation in the transverse plane.

It seems that the transient internal rotational movement may reduce the contrary vertical rotation generated by the swing leg and makes the free moment of the latter part of the oblique foot smaller than the rigid foot.

However, this hypothesis still needs detailed research to confirm.

- Comment 2:

About some technical details missing:

1. Fig 10, x-axis label: It is unclear what "Load" refers to.

Thanks for mentioning that this is missing.

We changed the "Load" to "AL" (Axial Loading).

We believe this label may help the reader intuitively associate AL to axial loading

The change is shown below.

Figure 10. Average values of free moment generation under 150 N axial load.(The magnitude is normalized by the axial load (AL).)

2. In Table 2, the origin with respect to which the CoP location is measured is not reported. The authors should report the location of the center of pressure with respect to a consistent landmark on the foot (say, the heel contact point) so that the relative location of the CoP can be interpreted between the different feet.

We added the heel contact points information of each foot in the revised draft.

Also, we found the mislabeling and revised the caption of Table 2.

The change is shown below.

5. Appendix

(a) Supplementary experimental data

The center of pressure (CoP) positions and heel contact point (HCP) of each test foot of the axial loading experiment is listed in Table 2. The HCP represents the consistent landmark of the foot. It is used to show the origin with respect to the CoP locations.

Table 2. The center of pressure (CoP) measurement of the axial loading experiment

Condition	CoP-X (mm)	CoP-Y (mm)	HCP-X (mm)	HCP-Y (mm)
Non-oblique foot	-42.4 ± 0.1	12.4 ± 0.8	-0.12 ± 0.04	-0.02 ± 0.04
Oblique foot	-20.6 ± 0.1	-7.1 ± 0.5	0.22 ± 0.16	0.1 ± 0.07
Rigid foot	-35.8 ± 0.2	-7.9 ± 2.4	0.18 ± 0.08	0.16 ± 0.09